# Comparison of Vegetables of Ecological and Commercial Production: Physicochemical and Antioxidant Properties

Zacnicté Olguín-Hernández [1], Quinatzin Yadira Zafra-Rojas [1,*], Nelly del Socorro Cruz-Cansino [1], Jose Alberto Ariza-Ortega [1], Javier Añorve-Morga [2], Deyanira Ojeda-Ramírez [3], Reyna Nallely Falfan-Cortes [2], Jose Arias-Rico [4] and Esther Ramírez-Moreno [1,*]

[1] Área Académica de Nutrición, Centro de Investigación Interdisciplinario, Instituto de Ciencias de la Salud, Circuito Actopan-Tilcuautla s/n. Ex-hacienda La Concepción, San Agustín Tlaxiaca 42160, Hidalgo, Mexico; olguinh@uaeh.edu.mx (Z.O.-H.)

[2] Área Académica de Química, Instituto de Ciencias Básicas e Ingeniería, Ciudad del Conocimiento, Carretera Pachuca-Tulancingo Km. 4.5 Col. Carboneras, Mineral de la Reforma 42184, Hidalgo, Mexico

[3] Área Académica de Medicina Veterinaria y Zootecnia, Instituto de Ciencias Agropecuarias, Universidad Autónoma del Estado de Hidalgo, Av. Universidad Km 1, Ex-Hda. de Aquetzalpa, Tulancingo 43600, Hidalgo, Mexico

[4] Área Académica de Enfermería, Centro de Investigación Interdisciplinario, Instituto de Ciencias de la Salud, Circuito Actopan-Tilcuautla s/n. Ex-hacienda La Concepción, San Agustín Tlaxiaca 42160, Hidalgo, Mexico

\* Correspondence: quinatzin_zafra@uaeh.edu.mx (Q.Y.Z.-R.); esther_ramirez@uaeh.edu.mx (E.R.-M.)

**Abstract:** This research aimed to compare some physicochemical and antioxidant properties in vegetables (chard, beet, coriander, spinach, lettuce, radish, carrot, and tomato) of ecological and commercial production. The ecological products were cultivated and obtained from three harvests in an ecology garden with standardized methodologies for implementation while the commercial samples were obtained from a local supplier. On the same purchase or harvest day, the color, texture, moisture, and ashes parameters were measured in the fresh produce without unpeeling. In the lyophilized samples, bioactive compounds (total phenolic compounds, ascorbic acid, chlorophyll *a* and *b*, β-carotenes, anthocyanins, betalains, and lycopene) were determined, and antioxidant activity was found using the 2,2-azino-bis-3-ethylbenzothiazoline-6-sulfonic acid (ABTS$^{\bullet+}$), 2,2′-diphenyl-1-picrylhydrazyl (DPPH) and ferric-reducing antioxidant power (FRAP) assays, and chelating activity. The ecological vegetables presented better color (high luminosity and intensity) than commercial samples, and, according to the value of ΔE, this is a difference that can be perceived by the human eye. In the same way, the ecological vegetables were more turgid than the commercial samples ($p < 0.05$). The content of bioactive compounds was found in higher concentrations in ecologically produced vegetables and this was correlated positively with antioxidant capacity. It is important to carry out more studies to determine the effect on health of these vegetables when they are integrated into the diet and thus to be able to recommend their inclusion in the diet as a sustainability strategy in the production of vegetables for self-consumption.

**Keywords:** ecological vegetables; phenolic; bioactive compounds; antioxidant activity

## 1. Introduction

A food security system is based on economic, social, and environmental factors to provide safe food for the entire population. In addition, it is completely profitable, with benefits for society, and has a positive or neutral impact on the environment (economic, social, and environmental sustainability) [1]. Therefore, a food security system requires carefully diversified food sources in terms of land (diversification of crops, organic fertilizers) and environmental sustainability (production of food in a way that does not affect the environment) to satisfy the nutritional requirements of the people [2]. The consumption of vegetable food can play an important role in improving the nutritional status of the



population and preventing non-communicable and deficiency diseases, among others, due to its well-recognized nutritional and medicinal value [3,4]. Starting in the year 2000, certification processes for food production were established in the United States with methodologies that respect ecosystems and that do not use fertilizers or pesticides that compromise the environment; this process is called "organic production" which implies administrative procedures and an economical cost that the producer absorbs [5]. Consequently, the concept of organic farming was implemented in agricultural systems centered on the need to develop technologies and practices that do not have adverse effects on the environment (goods and services), but that are accessible and effective for farmers, which leads to improvements in food productivity. In this process, it may be understood that everyone follows the standards of an organic system but with the term of "ecological production" [6]. This ecological process does not have a certification; it is only validated by the ethical practices of the responsible producer [7].

A sustainability strategy is the production of food for self-consumption, using vertical orchards, green roofs, and ecological school gardens when feasible; vegetables for frequent consumption can be produced in them, allowing access to these vegetable products all the time [8,9]. These strategies take care of the environment, avoid food waste through compost production, contribute to the economy of families, and provide better food quality (higher bioactive compounds, dietary fiber, minerals, and others) [10].

This food production system turns out to be more attractive to consumers [11]. It can also contribute to the diminished food supply chain and storage time, and, therefore, preserve the hygienic and nutritional qualities of the vegetables, including those that are perceived by the senses (taste, smell, color, texture, shape, and appearance) related to freshness and better flavor [12,13].

The physical characteristics of vegetables are the result of the chemical properties of the plant and are associated with the stability of their cell membranes, the distribution of moisture in the tissues, and the synthesis and integrity of bioactive compounds such as phenolic compounds, vitamins, minerals, and pigments [14]. These have been associated with biological functions in the human body that favor health effects, especially as antioxidants [15].

Consequently, vegetables that are produced using organic methods may have higher concentrations of these compounds [16]. The synthesis of these compounds occurs when the plants are exposed to controlled stress conditions (as occurs in organic products where mineralization or inorganic fertilization process is not carried out) [17,18]. So, the vegetables that are produced in this way may have a greater presence of these compounds in regional production. As a consequence, these protection mechanisms promote an increase in bioactive compounds during plant growth in ecological production systems. If the handling and storage conditions (time, humidity, temperature, and pH) are controlled during postharvest, the tissues of the plants are affected, which will modify the concentration of the bioactive compounds due to losses or modification of their natural form. For this reason, variations in production systems and storage conditions could explain the differences in pigment concentration and humidity between ecological and commercial vegetables [19].

Considering how production systems influence the physical and chemical quality of vegetables, and the little information there is on the quality of vegetables that are produced in an ecological system, this study aims to compare some physical properties (color and texture) and antioxidant properties in vegetables in an ecological garden against commercial products.

## 2. Materials and Methods

### 2.1. Obtaining Vegetable Material

The commercial samples were obtained from a local supplier of Pachuca Hidalgo, México, choosing the color and texture by considering commercial maturity without lacerations or external lesions, as would be selected by a local consumer. The ecological production was obtained in the same city, with a dry temperate climate and an annual

average of 15 °C with winds blowing from the northeast throughout the year [20]. It was carried out as follows: Homemade compost was generated, which was left to mature; the sowing substrate prepared with compost and plant soil (50:50) was placed in wooden containers, and 6 plants per container were sown; irrigation was performed as needed. Pest management was done culturally (garlic, chili and pepper as natural insecticides for aphids and whiteflies). The harvest was carried out according to the maturity of the vegetable. For the analysis, greens from 3 harvests in different ecological gardens were used. The analyzed species were chard (*Beta vulgaris var. cicla*), beetroot (*Beta vulgaris*), coriander (*Coriandrum sativum*), spinach (*Spinacia oleracea*), lettuce (*Lactuca sativa*), radish (*Raphanus sativus*), carrot (*Daucus carota sativus*), and tomato (*Solanum*). These vegetables were analyzed between September 2021 and February 2022 based on their natural maturity, and all the plant materials were obtained according to their similar sensory characteristics (appearance, color, and texture), including the commercial samples. All the samples were carried to the laboratory in a thermal container. The color, texture, and moisture parameters were measured in the fresh produce without unpeeling the same purchase or harvest day. In another part of the vegetables, the edible parts of the vegetables (20–50 g) were cut into small pieces (<0.5 cm) and lyophilized (LABCONO WWR26671-581, Kansas City, MO, USA) at $-52$ °C for 48 to 72 h. Lyophilisates samples were ground (analytical mill A11 2,900,001, EE. UU.), homogenized, and sieved into particles less than 500 μm. The powder was stored in a resealable plastic bag until analysis.

### 2.2. Physical Properties

2.2.1. Color Measurement

The color was recorded with a Chroma meter CR-300 series (CE Minolta, Osaka, Japan). In the leaves, color was measured at three points and on the back. In the vegetables of beet, tomato, radish, and carrot, two external points and one internal were taken. The CIE-Lab parameters were determined by luminosity ($L^*$), $a^*$, and $b^*$ coordinates.

The data of $L^*$, $a^*$, and $b^*$ were converted to color difference ($\Delta E$) using the following equation [21]:

$$\Delta E = \sqrt{(\Delta L^*)^2 + (\Delta a^*)^2 + (\Delta b^*)^2} \tag{1}$$

$\Delta E$: Color difference;
$\Delta L^{*2} = (L^*_1 - L^*_2)^2$ where $L^*_1$ is commercial vegetable and $L^*_2$ is ecological vegetable
$\Delta a^{*2} = (a^*_1 - a^*_2)^2$ where $a^*_1$ is commercial vegetable and $a^*_2$ is ecological vegetable
$\Delta b^{*2} = (b^*_1 - b^*_2)^2$ where $b^*_1$ is commercial vegetable and $b^*_2$ ecological vegetable

2.2.2. Texture, Moisture, and Ashes

The parameters of texture to assess turgor in vegetables were determined using a texture analyzer (Texture Analyzer, TA Plus, Stable Microsystems Co., Surrey, UK). A Kramer shear cell with a five-blade probe was used to analyze leafy vegetables, performed in 10 replicates per sample (placing 20 g of chopped vegetables in a load cell for a Kramer probe with 5 cutting blades). For the carrot, radish, beet, and tomato, a puncture probe P/2N was used to assess the skin breaking strength and the pulp firmness. The test was performed at a 10 mm distance using a test speed of 2 mm/s of force. The sample was analyzed and reported as maximum force (N/g) [22].

The moisture was determined according to AOAC procedure 945.15 by desiccation at 105 °C until constant weight and the ashes were obtained according to AOAC procedure 962.09.

### 2.3. Bioactive Compounds Analysis

2.3.1. Extraction of Antioxidant Compounds

To carry out the extraction process, 250 mg of lyophilized samples was weighed and extracted by shaking at room temperature with 50 mL of a mixture of methanol–water (50:50 $v/v$, 60 min, room temperature; constant shaking) and 50 mL of mixture of acetone–

water (70:30 $v/v$, 60 min, room temperature; constant shaking) [23]. After centrifugation (15 min, 25 °C, 3000 rpm), the supernatants were brought to a total volume of to 25 mL, with a mixture (1:1) of the solution of solvents and used to determine total phenolic compounds, ascorbic acid, anthocyanin, betalains, and antioxidant activity (ABTS$^{\bullet+}$, FRAP, DPPH, and chelating activity of ferrous ions).

### 2.3.2. Total Phenolic Compounds

The total phenolic content was estimated based on the Folin–Ciocalteau procedure [24]. An aliquot of the extract solution (100 μL) was mixed with 0.5 mL of Folin–Ciocalteu reagent (2.5 mL, previously diluted with water 1:10 $v/v$) and sodium carbonate (75 g/L). After adding 400 μL of sodium carbonate (7.5%), samples were allowed to stand for 30 min at room temperature, and the absorbance was read at 765 nm using a microplate reader (Power Wave XS UV-Biotek, software KC Junior, Winooski, VT, USA). The gallic acid was used as a reference standard, and the results were expressed as milligrams of gallic acid equivalents per 100 g of dry weight (mg GAE/100 g dw).

### 2.3.3. Ascorbic Acid

The ascorbic acid content in the vegetable extract was determined according to Dürüst et al. [25]. The sample was diluted 1:10 in a solution of 0.4% oxalic acid. Briefly, 100 μL of the extract was mixed with 100 μL of acetate buffer and 800 μL of 2,6-dichlorophenolindophenol (DCPI). The absorbance of the mixture was measured at 520 nm in the microplate reader (Power Wave XS UV-Biotek, software KC Junior, USA), and ascorbic acid was used as a reference standard, and the results were expressed as mg ascorbic acid per 100 g of dry weight (mg AA/100 g dw).

### 2.3.4. Total Anthocyanins Content (TAC)

The spectrophotometric pH differential method mentioned in the literature [26] was employed to measure the total anthocyanin content (TAC) in the extracts. Two dilutions of the same sample were made in 0.025 M potassium chloride pH 1.0 and 0.4 M sodium acetate pH 4.5. The pH was adjusted by adding concentrated HCl respectively. The absorbance of each dilution was measured a 520 and 700 nm using a Power Wave XS UV-Biotek, software KC Junior, USA. The total anthocyanin content (TAC) was determined as follows:

$$Abs = (Abs_{510} - Abs_{700})pH_{1.0} - (Abs_{510} - Abs_{700})pH_{4.5}$$

$$TAC \left(\frac{\text{mg}}{\text{L}}\right) = \frac{Abs * Mw * DF * 1000}{\varepsilon * 0.52} \tag{2}$$

where:

*Abs*: Absorbance
*Mw:* Molecular weight of cyanidin-3-O-glucoside (449.2 g/mol).
*DF*: Dilution factor of the sample.
*ε:* Molar extinction coefficient of cyanidin-3-O-glucoside 26,900 L mol$^{-1}$ cm$^{-1}$.
0.52 cm: Path length

### 2.3.5. Total Betalain Content (TBC)

The total betalain content (TBC) of the betacyanins and betaxanthins content was determined according to the method described by Fernández-López et al. [27] with some adaptations. An aliquot of 100 μL of the extract was determined by spectrophotometer UV/VIS (Power Wave XS UV-Biotek, software KC Junior, USA) at 480 nm for betacyanin and 535 nm for betaxanthin [28]. The betalain content was determined as follows and the results were expressed in mg.

$$TBC \left(\frac{\text{mg}}{\text{g}}\right) = \frac{A * FD * MM * 1000}{E * 0.29} \tag{3}$$

*A*: Absorbance
*FD*: Dilution factor
0.29 mL: Tucket length
*E*: Molar extinction coefficient
The following values were used:
*MM* = 550 g/mol, $\varepsilon$ = 60.0 L/mol cm in $H_2O$ for betacyanin determination
*MM* = 308 g/mol, $\varepsilon$ = 48.0 g/mol cm of $H_2O$ for betaxanthin determination

### 2.3.6. β-Carotene, Chlorophylls, and Lycopene

Extraction Method

To the determination of pigments, 500 mg of the lyophilized sample was homogenized with 10 mL of an acetone–hexane (2:3) mixture for 2 min. Immediately, the samples were sonicated (sonicator Bandelin HD3100 Sonopuls, Hamburg, Germany) for 3 min (5 cycles: Puls 30 s, pause 10 s). Homogenates were centrifuged (Eppendorf Hamburg, Germany) at 5000 rpm for 10 min at 20 °C. The sample was frozen until used and it was maintained in an ice-water bath to prevent overheating [29].

The pigments were determined according to the method of Nagata and Yamashita [29]. The absorbance spectrum of each supernatant was measured and the absorption maxima were read at 453, 505, 645, and 663 nm (Power Wave XS UV-Biotek, software KC Junior, EE. UU.). The β-carotene and chlorophyll (Chl *a*, Chl *b*) content were calculated (Microsoft Office Excel 97-2003) from the following equations:

$$\beta - carotene\left(\frac{mg}{100\text{ mL}}\right) = (0.216 * A663) - (1.22 * A645) - (0.304 * A505) + (0.452 * A453) \tag{4}$$

$$Chlorophyll\ a\left(\frac{mg}{100\text{ mL}}\right) = (0.999 * A663) - (0.0989 * A645) \tag{5}$$

$$Chlorophyll\ b\left(\frac{mg}{100\text{ mL}}\right) = (0.328 * A663) + (1.77 * A645) \tag{6}$$

The lycopene was determined according to the method of Nagata and Yamashita [30]. The determination of lycopene was performed only on tomato samples. The absorbance spectrum of the extract of each supernatant was measured, and the absorption maxima was read at 453, 505, 645, and 663 nm (Power Wave XS UV-Biotek, software KC Junior, EE. UU.). The lycopene content was calculated from the following equation:

$$Lycopene\left(\frac{mg}{100\text{ mL}}\right) = (0.0485 * A663) - (0.204 * A645) + (0.372 * A505) - (0.0806 * A453) \tag{7}$$

### 2.4. Antioxidant Activity

2.4.1. 2,2-Azino-bis-3-ethylbenzothiazoline-6-sulfonic Acid (ABTS•+)

The antiradical capacity was measured according to Kuskoski et al. [31]. Briefly, the radical cation (ABTS•+) was produced by reacting 7 mmol/L ABTS•+ stock solution with 2.45 mmol/L potassium persulfate under dark conditions and room temperature for 16 h before use. The ABTS•+ solution was diluted with deionized water to an absorbance of 0.70 ± 0.10 at 754 nm. Then, 20 μL of the extract of the sample was added to 980 μL of diluted ABTS•+ solution; absorbance readings (754 nm) were taken in a microplate reader (Power Wave XS UV-Biotek, software KC Junior, EE. UU.) after incubation for 7 min at room temperature. The antioxidant capacity was expressed as micromole of Trolox equivalents per 100 g of dry weight (μmol TE/100 g dw).

2.4.2. 2,2′-Diphenyl-1-picrylhydrazyl (DPPH)

The antiradical activity was measured with 1,1-diphenyl-2-picrylhydrazylradical (DPPH) [30]. An ethanolic solution (7.4 mg/100 mL) of the stable DPPH radical was prepared. Then, the extracts (100 μL) were taken into vials and 500 μL of DPPH solution were added, and the mixture was for to stand 1 h at room temperature. Finally, absorbance was measured at 520 nm using a microplate reader (Power Wave XS UV-Biotek, software

KC Junior, Winooski, VT, USA). The standard curve was concentrations of 0, 50, 100, 200, and 300 μmol of Trolox. Free radical scavenging activity was expressed as micromole of Trolox equivalents per 100 g on a dry weight (μmol TE/100 g dw).

### 2.4.3. Ferric-Reducing Antioxidant Power (FRAP)

The FRAP method was assayed according to Benzie and Strain [31]. FRAP solution with 2,4,6-tripyridyl-s-triazine (TPTZ) 10 mM solution in 40 mM HCl, $FeCl_3$ 20 mM, and acetate buffer 300 mM (sodium acetate anhydrous and glacial acetic acid, pH 3.6). An aliquot of 30 μL of the sample was mixed with 90 μL of distilled water and 900 μL of FRAP solution in the dark room. Absorbance was measured at 593 nm using a microplate reader (Power Wave XS UV-Biotek, software KC Junior, EE. UU). The results were expressed as micromoles of Fe (II) per 100 g of dry weight (μmol Fe (II)/100 g dw) of each one of the vegetables.

### 2.4.4. The Chelating Activity of Ferrous Ions

The chelating activity (CA) was determined as described by Gulcin et al. [32]. Briefly, 100 μL of the undiluted sample was placed in vials and 50 μL of ferric (II) chloride solution (2 mM) and 450 μL of methanol were added. The solution was vortexed and left for 5 min at room temperature before adding 400 μL of ferrozine (5 mM). The mixture was vortexed and then allowed to settle for 10 min at room temperature. The absorbance was read at 562 nm in a microplate reader (Power Wave XS UV-Biotek, software KC Junior, EE. UU.) using EDTA (0.1 M) [33], and deionized water as control. The chelating activity was calculated using the following equation:

$$\% \, CA \frac{A0 - A1}{A0} * 100 \tag{8}$$

$A0$: Absorbance of the control;
$A1$: Absorbance of the sample.

### 2.5. Statistical Analysis

All data were reported as mean ± standard deviation for at least three repetitions in each treatment. To determine differences between samples of ecological and commercial vegetables to levels of statistical significance a Student *t*-test was conducted. In both statistical tests, the significance level was set at $p = 0.05$. The analysis of the correlation between the results of antioxidant activity and bioactive compounds, also between the color results and concentration of pigments, was carried out using the Spearman test. Data were analyzed with program SPSS (System for WINTM version 25, Institute Inc., Cary, NC, USA).

### 3. Result and Discussion

### 3.1. Physical Properties

### 3.1.1. Color

The color of foods is an important attribute and it is associated with consumer acceptability and preference [34]. Color measurements on the front and back of the leafy vegetables (chard, coriander, spinach, and lettuce) were performed, as well as in the pericarp and pulp of whole-piece vegetables (beet, tomato, radish, and carrot) (Tables 1 and 2, respectively). In the *L** parameter, the ecologically produced vegetables were significantly ($p < 0.05$) brighter than the commercial vegetables. This behavior occurred both in leafy vegetables and in those evaluated in the whole piece, except for the tomato sample in the external part (Table 1). Concerning the *a** and *b** coordinates, all green vegetables were in the green–yellow quadrant, while red vegetables were in the red–green quadrant, apart from the radish in the inside part (Table 2).

**Table 1.** Color of the adverse/external part of vegetables from the ecological garden and from commercial production.

| Vegetable | Type | L* | a* | b* | ΔE |
|---|---|---|---|---|---|
| Chard | Com | 36.32 ± 3.64 | −6.32 ± 0.81 | 46.32 ± 3.48 | 23.49 ± 2.20 |
| | Eco | 54.88 ± 3.62 * | −9.52 ± 0.81 * | 53.16 ± 4.32 * | |
| Coriander | Com | 34.60 ± 1.64 | −7.75 ± 0.88 | 48.82 ± 4.27 | 13.08 ± 0.92 |
| | Eco | 46.69 ± 3.52 * | −10.65 ± 0.75 * | 56.55 ± 3.03 * | |
| Spinach | Com | 30.25 ± 2.91 | −6.72 ± 1.25 | 42.45 ± 4.47 | 11.81 ± 0.79 |
| | Eco | 37.08 ± 2.58 * | −10.25 ± 0.42 * | 50.19 ± 3.18 * | |
| Lettuce | Com | 53.76 ± 3.70 | −5.46 ± 1.05 | 22.89 ± 2.21 * | 9.57 ± 0.32 |
| | Eco | 59.00 ± 5.32 * | −5.88 ± 0.81 | 21.37 ± 2.17 | |
| Beet | Com | 28.71 ± 2.87 | 5.58 ± 0.87 | 23.77 ± 2.24 | 12.91 ± 1.10 |
| | Eco | 34.73 ± 2.65 * | 16.77 ± 4.86 * | 28.15 ± 5.85 | |
| Carrot | Com | 48.64 ± 2.38 | 33.33 ± 1.36 | 59.99 ± 4.31 | 10.49 ± 1.09 |
| | Eco | 52.92 ± 2.48 * | 27.35 ± 3.75 * | 62.50 ± 3.77 | |
| Tomato | Com | 36.27 ± 1.54 | 23.72 ± 2.30 | 45.61 ± 1.21 | 2.79 ± 0.40 |
| | Eco | 34.48 ± 2.09 * | 26.97 ± 3.62 | 45.42 ± 5.32 | |
| Radish | Com | 29.07 ± 2.09 | 39.19 ± 3.61 | 24.98 ± 3.21 | 4.67 ± 0.03 |
| | Eco | 34.47 ± 1.25 * | 51.81 ± 2.74 * | 47.16 ± 1.98 * | |

COM: Commercial and ECO: Ecological. * significantly different at $p < 0.05$ in the two-sided test of equality for column means. Cells without a subscript are not included in the test. The tests assume equal variances.

**Table 2.** Color of the back/inside part of vegetables from the ecological garden and from commercial production.

| Vegetable | Type | *L** | *a** | *b** | ΔE |
|---|---|---|---|---|---|
| Chard | Com | 36.32 ± 2.46 | −6.32 ± 0.92 | 39.32 ± 2.24 | 5.7 ± 0.62 |
| | Eco | 44.50 ± 2.87 * | −9.91 ± 0.63 * | 48.09 ± 2.67 * | |
| Coriander | Com | 33.24 ± 3.20 | −8.60 ± 0.54 | 40.50 ± 4.03 | 8.64 ± 0.19 |
| | Eco | 47.94 ± 3.85 * | −10.14 ± 0.26 * | 45.75 ± 4.52 * | |
| Spinach | Com | 37.72 ± 2.21 | −7.15 ± 0.96 | 47.78 ± 3.78 | 12.70 ± 0.27 |
| | Eco | 44.97 ± 3.08 * | −8.39 ± 5.71 | 50.34 ± 4.82 | |
| Lettuce | Com | 61.16 ± 4.76 | −7.00 ± 1.15 | 33.28 ± 3.40 * | 8.76 ± 0.69 |
| | Eco | 65.33 ± 3.47 * | −7.00 ± 0.66 | 27.19 ± 2.63 | |
| Beet | Com | 11.25 ± 0.75 | 36.58 ± 2.59 | 22.53 ± 3.88 | 4.63 ± 0.40 |
| | Eco | 15.09 ± 1.44 * | 38.92 ± 1.69 | 23.14 ± 2.38 | |
| Carrot | Com | 53.80 ± 3.80 | 34.00 ± 5.67 | 45.91 ± 2.96 | 12.7 ± 1.06 |
| | Eco | 59.06 ± 2.24 * | 35.85 ± 2.32 | 58.95 ± 6.24 * | |
| Tomato | Com | 50.15 ± 6.50 | 15.77 ± 2.79 | 23.27 ± 2.60 | 8.56 ± 0.67 |
| | Eco | 55.11 ± 9.90 | 17.68 ± 1.20 * | 23.19 ± 2.64 | |
| Radish | Com | 74.36 ± 2.36 | 0.59 ± 0.21 | 6.66 ± 0.97 | 2.8 ± 0.49 |
| | Eco | 75.77 ± 1.57 | −0.44 ± 0.14 * | 7.92 ± 0.67 * | |

COM: Commercial and ECO: Ecological. * significantly different at $p < 0.05$ in the two-sided test of equality for column means. Cells without a subscript are not included in the test. The tests assume equal variances.

All green vegetables in ecological production were significantly different in the parameter *a** (green–red) compared to commercial samples, showing a greener color trend,

excluding the lettuce (ecological and commercial samples), which had similar values in both parts of the studied vegetables, while the spinach was similar between the vegetable samples only at the rear. Regarding red vegetables, the ecological samples showed a tendency to be red, placing the values in the red–yellow quadrant; these samples did not present significant differences, and only the commercial carrot in the external value of *a\** was significantly higher with a tendency to be orange in color.

About the parameter *b\** (yellow–blue), the ecological vegetables such as chard, coriander, spinach, and radish had a higher yellow color than commercial products ($p < 0.05$), omitting lettuce in the commercial vegetables (Table 2).

The ΔE value indicates the perception of color difference by the human eye and according to Simunovic [35], values greater than 3 indicate that the color differences are perceptible to the naked eye, and the consumer associates this difference in color with greater freshness [36]. The ΔE values at all points of the evaluated vegetables in this study were higher than 3, ranging from 4.67 to 23.49, (Tables 1 and 2), without counting the outer part of the tomato and the internal part of the radish, which have values slightly less than 3 (2.79 and 2.8 respectively).

### 3.1.2. Texture, Moisture, and Ashes

The results of texture, moisture, and ashes are shown in Table 3. It can be observed that the texture reported less shear force in the vegetables from the ecological garden, which is associated with better turgidity. This is an indicator of freshness and quality in vegetables because it is maintained when water is found in plant tissues, in the correct compartments such as vacuoles, cytosol, and intermembrane space [32,37].

**Table 3.** Texture, moisture, and ashes of vegetables from the ecological garden and from commercial production.

| Vegetable | Type | Texture (N/100 g o Piece) | Moisture (%) | Ashes (%) |
|---|---|---|---|---|
| Chard | Com | 3684 ± 312 | 88 ± 0.04 | 0.11 ± 0.001 |
| | Eco | 3146 ± 289 * | 91 ± 0.03 | 0.12 ± 0.001 |
| Coriander | Com | 4825 ± 228 | 87 ± 0.01 | 0.10 ± 0.01 |
| | Eco | 3537 ± 228 * | 87 ± 0.01 | 0.22 ± 0.01 * |
| Spinach | Com | 4884 ± 343 | 89 ± 0.01 * | 0.018 ± 0.03 |
| | Eco | 2878 ± 171 * | 84 ± 0.02 | 0.10 ±0.01 |
| Lettuce | Com | 2510 ± 241 | 93 ± 0.01 | 0.03 ± 0.00 |
| | Eco | 2210 ± 213 * | 91 ± 0.01 | 0.08 ± 0.02 * |
| Beet | Com | 15.85 ± 0.84 | 86 ± 0.02 | 0.08 ± 0.002 |
| | Eco | 15.08 ± 0.66 * | 85 ± 0.04 | 0.13 ± 0.023 |
| Carrot | Com | 18.99 ± 1.72 | 97 ± 0.45 | 0.09 ± 0.019 |
| | Eco | 16.51 ± 0.73 * | 89 ± 0.03 | 0.97 ± 0.017 |
| Tomato | Com | 1.66 ± 0.08 | 93 ± 0.0 | 0.02 ± 0.00 |
| | Eco | 0.60 ± 0.09 * | 94 ± 0.0 * | 0.06 ± 0.02 * |
| Radish | Com | 6.85 ± 0.70 | 94 ± 0.01 * | 0.03 ± 0.00 |
| | Eco | 3.69 ± 0.47 * | 91 ± 0.06 | 0.02 ± 0.01 |

COM: Commercial and ECO: Ecological. * significantly different at $p < 0.05$ in the two-sided test of equality for column means. Cells without a subscript are not included in the test. The tests assume equal variances.

Most of the samples did not show significant differences in moisture, except for commercial spinach and radish which had higher moisture, and the ecological tomato had higher moisture than the commercial ones. This behavior may be due to postharvest storage, handling conditions, and the sale period [37]. An usual commercial practice of the

merchants is to keep the vegetable products submerged in water or with constant spraying so that the vegetables maintain their fresh appearance [38]. The water in the vegetable is absorbed by the hemicellulose networks of plants; however, this water content is placed on the surface and not in the proper cell compartments [39]. This type of management was not carried out on ecological vegetables since these were harvested and immediately transferred for physical analysis. Because this process generates hydration of the hemicellulose network of the cell wall, increasing its elasticity, it can be verified with the evaluation of texture [40].

The ash values showed significant statistical differences in ecological coriander, tomato, and lettuce in comparison with commercial vegetables. According to Butnariu et al. [41], edible vegetables contain minerals, mainly K, Ca, Mg, P, and Fe, and oligo-element traces (Cr, Cu, I, F, Zn, Mg, Mo, and Se). The mineral and trace element content of plants is known to be affected by the cultivar of the plant, soil, weather conditions, use of fertilizers, and by the state of the plant maturity during the growing season and after the harvest [42].

### 3.2. Bioactive Compounds

Antioxidant compounds have an essential role in human health due to these compounds preventing different types of chemical damage caused by free radicals, avoiding cell oxidation, and therefore reducing the risk of non-communicable diseases [43,44].

In general, all ecologically produced vegetables showed a higher concentration of antioxidant compounds than commercial samples as presented below.

### 3.2.1. Total Phenolics Compounds (TPC)

The vegetables of ecological production had a concentration of TPC between 553 to 1429 mg GAE/100 g dw, and the samples with a high concentration were coriander (1429.35 mg GAE/100 g dw) and radish (1120.26 mg GAE/100 g dw), while the carrot presented the lowest concentration (102.22 mg GAE/100 g dw). Specifically, the TPC concentration in ecological vegetable products was significantly higher ($p < 0.05$) than in commercial products, except in tomatoes (Table 4). According to Oliveira [45], the synthesis of these compounds occurs when the plants are exposed to controlled stress conditions (as occurs in organic products where mineralization or inorganic fertilization process is not carried out), or during ripening, especially with a major activity (up to 140%) of phenylalanine ammonia-lyase (PAL) enzyme activity caused by physiological alterations [46].

### 3.2.2. Ascorbic Acid

The studied vegetables had a content of ascorbic acid between 932 to 1310 mg AA/100 g dw; however, the ecological vegetables with a high concentration were chard, coriander, tomato, lettuce, and radish. The lettuce of ecological production was the vegetable with the highest concentration of ascorbic acid (1310 mg AA/100 g dw), followed by chard with 1276.51 mg AA/100 g dw (Table 4). Considering that these vegetables in their fresh form contribute on average of 12.18 mg/100 g fw (fresh weight) of ascorbic acid, this amount could cover more than 100% of the recommended RDI (75 mg/day) for ascorbic acid in adults [47]. The vegetable with the lowest concentration of ascorbic acid was the commercial tomato with 940.16 mg AA/100 g dw. It is possible that the low availability of nitrogen in organic or ecological products decreases the synthesis of biomass in the vegetable, but increases the concentration of ascorbic acid [48], as acclaimed by the behavior observed in vegetables such as spinach, tomato, lettuce, and radish according to other studies [46,49].

**Table 4.** Total phenolic compounds, ascorbic acid, β-carotene, chlorophyll *a* and *b* in vegetables from the ecological garden and from commercial production (mg/100 g dw).

| Vegetable | Type | TPC (GAE) | Ascobic Acid (AA) | β-Carotene | Chlorophyll *a* | Chlorophyll *b* |
|---|---|---|---|---|---|---|
| Chard | Com | 426.76 ± 6.68 | 1219.37 ± 7.27 | 50.68 ± 0.27 | 509.24 ± 8.21 | 1286.05 ± 8.27 |
| | Eco | 485.25 ± 2.38 * | 1276.51 ± 7.27 * | 50.34 ± 0.13 | 525.10 ± 3.06 * | 1302.07 ± 0.81 * |
| Coriander | Com | 1151.11 ± 27.74 | 1035.24 ± 8.25 | 50.71 ± 0.08 | 524.33 ± 2.39 | 1233.34 ± 1.94 |
| | Eco | 1429.35 ± 10.61 * | 1070.16 ± 7.21 * | 50.61 ± 0.20 | 529.14 ± 0.83 * | 1306.03 ± 1.31 * |
| Spinach | Com | 443.73 ± 14.82 | 1208.41 ± 40.50 | 49.21 ± 0.29 | 508.18 ± 3.03 | 1261.31 ± 3.19 |
| | Eco | 592.50 ± 5.46 * | 1222.71 ± 90.02 | 49.47 ± 0.14 | 508.48 ± 6.92 | 1271.62 ± 8.08 |
| Lettuce | Com | 205.93 ± 3.67 | 1262.38 ± 17.17 | 11.61 ± 0.29 | 79.57 ± 5.53 | 128.39 ± 4.69 |
| | Eco | 754.23 ± 44.41 * | 1310.00 ± 12.60 * | 47.20 ± 0.28 * | 508.64 ± 3.29 * | 854.94 ± 12.76 * |
| Beet | Com | 277.99 ± 8.29 | 1284.44 ± 7.27 * | 1.44 ± 0.04 | 4.47 ± 0.36 | 11.08 ± 0.86 |
| | Eco | 553.15 ± 33.22 * | 1043.17 ± 11.68 | 2.55 ± 0.16 * | 44.13 ± 0.27 * | 68.44 ± 1.65 * |
| Carrot | Com | 88.18 ± 7.77 | 1278.25 ± 13.75 | 52.05 ± 0.09 | 12.71 ± 2.40 | 34.7 ± 1.65 |
| | Eco | 102.22 ± 3.96 * | 1310 ± 26.51 | 53.23 ± 0.04 * | 40.05 ± 5.38 * | 140.16 ± 5.11 * |
| Tomato | Com | 873.59 ± 2.56 * | 940.16 ± 11.98 | 18.04 ± 0.04 | 6.53 ± 0.30 | 21.49 ± 1.01 |
| | Eco | 506.89 ± 1.31 | 1038.57 ± 20.76 * | 19.33 ± 0.16 * | 12.61 ± 0.53 * | 13.39 ± 2.71 * |
| Radish | Com | 596.64 ± 4.26 | 932.22 ± 11.00 | 14.69 ± 0.59 | 19.99 ± 0.91 | 46.46 ± 0.45 |
| | Eco | 1120.26 ± 7.57 * | 1105.24 ± 19.05 * | 16.71 ± 0.19 * | 21.45 ± 0.16 | 45.74 ± 0.80 |

COM: Commercial and ECO: Ecological. * significantly different at *p* < 0.05 in the two-sided test of equality for column means. Cells without a subscript are not included in the test. The tests assume equal variances.

### 3.2.3. β-Carotene and Chlorophyll

The concentration of β-carotene in studied samples was between 47 to 53 mg/100 g dw, except for beets, radishes, and commercial lettuce which had lower values of this parameter. The results showed that the ecological production had a higher concentration of β-carotene than commercial samples, mainly in carrots, radishes, lettuce, tomatoes, and beets (Table 4). Considering fresh weight in the samples (1.52 to 5.04 mg/100 g fw), this concentration of β-carotene covers more the 50% of the RDI (2–4 mg/day) in the diet of an adult [50], with the exception of beet which only covers 10% of these recommendations. The lower values were in tomato, radish, and commercial lettuce (between 11.61 to 19.33 mg/100 g dw) and beet (1.44 to 2.55 mg/100 g dw). This difference in β-carotene concentration can be explained by the production methods (ecological and commercial) and the maturity of the plant at the time of harvest [51], but there is more scientific evidence that handling and postharvest storage are determining factors in the concentration of these compounds [52]. After 3 or 4 days postharvest, the loss of bioactive compounds becomes faster if the humidity and temperature conditions in storage are not adequate [53]. This could be the factor that is decisively influencing the results of the analysis carried out in this study.

Chlorophyll is a compound characteristic of green vegetables and it can be found in different chemical structures that contribute to several tones of green [54]. The content of chlorophylls in all the samples of the ecological type were higher than in commercial vegetables. The highest concentration of chlorophyll *b* (chemical structure leads to colors yellow-green) was observed in chard, coriander, spinach, and lettuce (1306.03 to 1271.62 mg/100 g dw), and therefore, chlorophyll *b* (blue-green) was lower in these products (508–524 mg/100 g dw) (Table 4).

### 3.2.4. Betalains, Lycopene, and Anthocyanins

The presence of anthocyanins in radishes, betalains in beets, and lycopene in tomatoes is characteristic and contributes to the color attributes of each vegetable, and it is associated

with their freshness [55,56]. The concentration of anthocyanins in ecological vegetables was 604.39 mg in radish, 5114 mg of betacyanins, and betaxanthins of 5567 mg in beet, and lycopene 2927 mg in tomato considering 100 g of dry mass while the commercial vegetables had of 25 to 48% lower concentration (Figure 1). Statistically, an association was found with color parameters (*b*\* and *a*\* coordinate) and concentration of these pigments in ecological vegetables ($r^2$ = 0.6 to 0.9\*). Also, there was a correlation between color and pigments in commercial vegetable samples, but with lower correlation values ($r^2$ = 0.4 to 0.6\*).

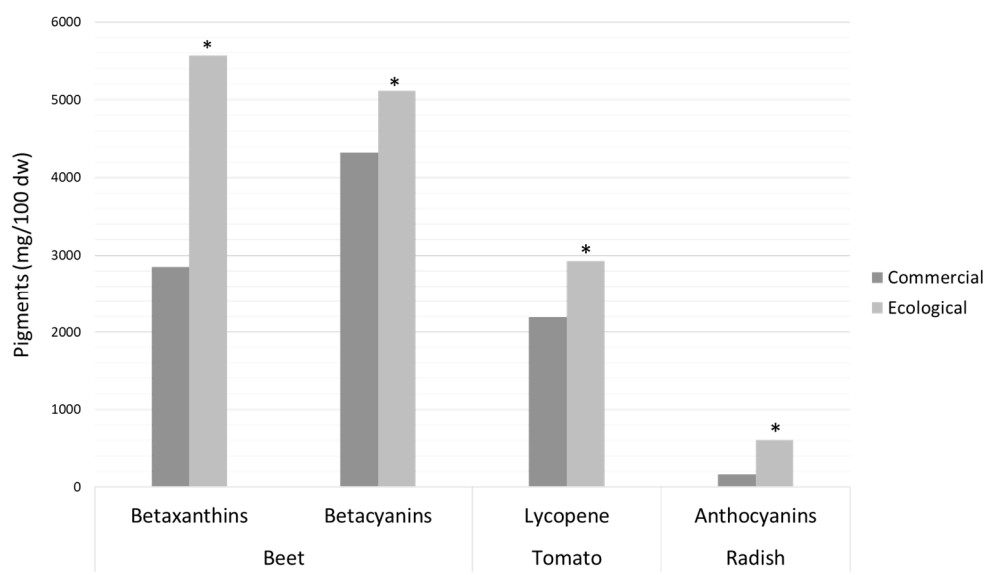

**Figure 1.** Concentrations of betalains in beet, lycopene in tomato, and anthocyanins in radish. \* The asterisk expresses the statistically significant difference at *p* < 0.05 in the two-sided test of equality between the commercial and ecological samples.

In general, commercial products could have more manipulations during processing, for instance practices such as washing and soaking [51,57]. Therefore, due to the high solubility of anthocyanins and betalains in water, the losses of these compounds could significantly increase in radishes and beets during storage and commercial distribution [53,54].

In particular, it has been observed that lycopene is higher in tomatoes that ripen on the plant in open environments, while in tomatoes produced in a greenhouse, lycopene synthesis could be higher during postharvest [46].

### 3.3. Antioxidant Activity

The presence of these compounds is related to antimicrobial, antiproliferative, anti-inflammatory, and antioxidant bioactivities [58,59]. Many postharvest conditions such as harvesting techniques, storage conditions, temperature, atmosphere, and pH could influence oxidative stress [60]. Some symptoms of oxidative stress in vegetables are inhibition of chloroplast development, core browning, superficial scald, disruption in membrane integrity, inactivation of protein due to the action of proteases, and bleaching of pigmentation [61]. Therefore, the presence of bioactive compounds contributes to the reduction of the oxidative state and the quality of vegetable products.

The Table 5 shows the results of antioxidant activity and chelating activity. The values of antioxidant activity measured as ABTS$^{\bullet+}$ were between 4137.38 to 6565.53 µmol TE/100 g dw. The samples of tomato and carrot had the lowest values (between 23 to 120.12 µmol TE/100 g dw) in ecological and commercial vegetables. In this study, this methodology reported high values of antioxidant activity compared to DPPH, FRAP, and chelating activity in vegetables, and it can be explained by the affinity with hydrophilic (polyphenols and ascorbic acid) and lipophilic (β-carotene, chlorophyll, and lycopene) compounds. A moderate association ($r^2$ = 0.567\*) was found between the TPC concentration and the

antioxidant activity measured by ABTS$^{\bullet+}$ in the vegetables from the ecological garden. These characteristics of the ABTS$^{\bullet+}$ method makes it possible to record the antioxidant activity of various compounds in the samples [62].

**Table 5.** Antioxidant activity, ascorbic acid, and iron chelating activity of vegetables from the ecological garden and from commercial production (100 g dw).

| Vegetable | Type | ABTS$^{\bullet+}$ (µmol TE) | DPPH (µmol TE) | FRAP (µmol Fe (II)) | Chelating Activity (%) |
|---|---|---|---|---|---|
| Chard | Com | 3961.95 ± 342.37 | 55.44 ± 3.11 | 1056.35 ± 1.24 | 85.57 ± 1.87 |
| | Eco | 3737.05 ± 159.54 | 108.52 ± 2.00 * | 3446.17 ± 207.77 * | 90.19 ± 0.03 * |
| Coriander | Com | 3596.63 ± 235.40 | 27.54 ± 1.07 | 1063.44 ± 374.00 | 79.34 ± 0.14 |
| | Eco | 3560.29 ± 226.21 | 73.00 ± 1.28 * | 1793.67 ± 15.48 * | 89.73 ± 0.78 * |
| Spinach | Com | 2465.32 ± 181.42 | 0.12 ± 0.00 | 124.03 ± 3.23 | 81.50 ± 0.42 |
| | Eco | 3395.38 ± 52.81 * | 0.14 ± 0.01 * | 176.94 ± 3.15 * | 92.75 ± 0.13 * |
| Lettuce | Com | 4951 ± 194 | 6390.2 ± 268.7 | 263.49 ± 1.22 | 21:01 ± 0.74 |
| | Eco | 6623 ± 639 * | 7600.32 ± 97.28 * | 865.05 ± 8.77 * | 49.56 ± 0.13 * |
| Beet | Com | 4362.63 ± 382.41 | 30.03 ± 2.78 | 197.49 ± 3.73 | 44.45 ± 1.00 |
| | Eco | 4231.84 ± 132.64 | 58.23 ± 1.93 * | 259.38 ± 6.12 * | 52.12 ± 0.09 * |
| Carrot | Com | 89 ± 6.01 | —– | 62.28 ± 4.61 | 23.86 ± 0.30 |
| | Eco | 120.12 ± 8.88 * | —– | 94.15 ± 3.49 * | 53.01 ± 4.50 * |
| Tomato | Com | 41.28 ± 1.62 * | 0.04 ± 0.0 | 37.99 ± 0.92 | 83.98 ± 0.15 |
| | Eco | 23.29 ± 1.17 | 0.07 ± 0.0 * | 57.27 ± 1.15 * | 92.35 ± 0.09 * |
| Radish | Com | 4137.38 ± 302.41 | 275 ± 14 | 135.79 ± 5.71 | 21.27 ± 0.85 |
| | Eco | 6565.53 ± 56.29 * | 1857 ± 101 * | 684.75 ± 13.79 * | 32.14 ± 2.56 * |

COM: Commercial and ECO: Ecological. * significantly different at *p* < 0.05 in the two-sided test of equality for column means. Cells without a subscript are not included in the test. The tests assume equal variances. —-: Not performed analysis.

The DPPH radical evaluates the free radical scavenging capacity of antioxidants according to their hydrogen-donating capacity. And it can evaluate the mechanisms of the transfer of hydrogen atoms and electrons [63]. As a result of this test, it can be observed that the vegetables produced in the ecological garden showed higher antioxidant activity than commercial samples. The ecological lettuce with 7600.32 µmol TE/100 g dw was the one that best captured free radicals (Table 5). On the other hand, in spinach and tomato, the values were lower than 0.07 µmol TE/100 g dw, while in carrots this assay was not detected. This activity was correlated with ascorbic acid concentrations with statistical significance ($r^2$ = 0.714*).

As in DPPH, in FRAP analysis the antioxidant activity registered by all samples of ecological vegetables was significantly (*p* < 0.05) higher than the commercially produced samples. The samples of ecological chard were the ones that reported the best antioxidant activity (3446.17 µmol Fe (II)/100 g dw) and the samples that recorded the lowest antioxidant activity were commercial tomatoes (37.99 µmol Fe (II)/100 g dw) (Table 5). In the vegetables from the ecological garden, the correlation was given from the values of the FRAP test with chlorophyll *b* ($r^2$ = 0.71*).

Table 5 also showed the results of the metal chelating assay. All the samples of the vegetables produced in the ecological garden have higher chelating activity than the samples of commercial production. The ecological spinach sample reported the highest percentage of chelation (92.35%), while commercial lettuce had the lowest value (21.01%). This result could be explained due to the strong iron-binding properties of polyphenols, whether the iron-chelating ability of catechol or polyphenols plays a key role in their antioxidant activity and anti-lipid peroxidation by blocking the Fenton reaction [64,65]. In addition, a moderate correlation was reported between the chelating assay with TPC content ($r^2$ = 0.657*).

## 4. Conclusions

The ecological production vegetables presented better physical characteristics in color (luminosity and intensity), which were perceptible by the human eye (ΔE). In addition,

they were more turgid compared to the commercial samples. Furthermore, the ecological products had higher bioactive compounds (TPC, ascorbic acid, β-carotenes, chlorophylls, anthocyanins, betalains, and lycopene) than the commercial products, and the content was positively correlated with antioxidant capacity.

However, it is essential to carry out more studies to clarify whether the differences found in favor of ecological products are reflected in the health of the people who consume them and thus promote the establishment of urban and peri-urban organic gardens as a strategy of sustainability for healthy eating.

**Author Contributions:** All the authors have contributed equally to this paper. All authors have read and agreed to the published version of the manuscript.

**Funding:** This research received no external funding.

**Data Availability Statement:** All data from the analyses performed are available from the corresponding authors of this publication.

**Conflicts of Interest:** The authors declare no conflict of interest.

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
