# Peer review of "Comparison of Vegetables of Ecological and Commercial Production: Physicochemical and Antioxidant Properties"

_sustainability, doi:10.3390/su15065117_

Round 1
Reviewer 1 Report
Dear authors,
I consider necessary improvements so that the article can be published.
Minor comments:
Line 16: Mexico(o).
Check spaces between words (Line 2, 17,...).
Line 97: replace "Sample" with "Obtaining plant material".
Line 417: do not put letters in bold (ABTS).
Line 419-429: unify font size.
Major comments:
Material and methods. Sections and subsections must be numbered. For example:
2.1. Obtaining plant material.
2.2. Physical properties
2.3. Bioactive compounds.
2.3.1. Extraction
...
2.3.2. Total phenols
...
2.4. antioxidant activity.
...
The cultivation techniques of both the local supplier and the organic garden must be described. The differences in the production process must be clear and detailed (type of soil, fertilization, irrigation, etc.).
The recommendations made by the authors as a conclusion should be rewritten (line 462-466). The fact that products of organic origin have quantitatively different properties and composition than non-organic products does not mean that they are healthier. To reach that conclusion, analysis would have to be done on the people who consume it. In addition, I would separate the fact of being better organoleptically and having a higher concentration of antioxidants from the claim of being more sustainable. Sustainability is linked to the reduction of inorganic inputs, water, resources, and this is not analyzed in this work.
(Sustainable =/= healthy =/= more color and more turgence)
Author Response
Response to Reviewer 1 Comments
We would like to thank reviewers for the comments made to our manuscript (ID: sustainability-2235770) entitled “Comparison of vegetables of ecological and commercial production: physicochemical and antioxidant properties”. Your contributions and suggestions have been considered and we include our responses point-by-point to the comments raised by the reviewers to improve the quality of the manuscript.
Point 1: Line 16: Mexico(o).
Answer: The letter “o” has been added in line 13 in the author data section.
Point 2: Check spaces between words (Line 2, 17,...).
Answer: The spaces have been added in full text
Point 3: Line 97: replace "Sample" with "Obtaining plant material".
Answer: The word has been replaced in line 105
Point 4: Line 417: do not put letters in bold (ABTS).
Answer: The “ABTS” in bold has been corrected without bold
Point 5: Line 419-429: unify font size.
Answer: The font size has been unified
Point 6: Material and methods. Sections and subsections must be numbered. For example:
2.1. Obtaining plant material.
2.2. Physical properties
2.3. Bioactive compounds.
2.3.1. Extraction
2.3.2. Total phenols
2.4. antioxidant activity.
Answer: The Materials and Methods section according to the reviewer suggestion has been reordered and numbered
Point 7: The cultivation techniques of both the local supplier and the organic garden must be described. The differences in the production process must be clear and detailed (type of soil, fertilization, irrigation, etc.).
Answer: We have included a paragraph explaining the cultivation techniques of ecological vegetable products, in the section on material and methods in lines 109-116. On the other hand, we do not have information on commercial samples because they were only bought in a retail market.
Point 8: The recommendations made by the authors as a conclusion should be rewritten (lines 462-466). The fact that products of organic origin have quantitatively different properties and composition than non-organic products does not mean that they are healthier. To reach that conclusion, the analysis would have to be done on the people who consume it. In addition, I would separate the fact of being better organoleptically and having a higher concentration of antioxidants from the claim of being more sustainable. Sustainability is linked to the reduction of inorganic inputs, water, and resources, and this is not analyzed in this work.
(Sustainable =/= healthy =/= more color and more turgence)
Answer: We have rewritten the conclusion indicating only the differences between organic and commercial vegetable products, lines 528-538. We consider that it is necessary to continue with studies to declare these products healthier. In addition, we have included the recommendation for the use of urban and periurban organic gardens as a sustainability strategy for healthy eating.
Reviewer 2 Report
Manuscript ID: sustainability-2235770
The article is very good and well described but there are some little mistakes :
Comment 1: Full forms of ABTS, DPPH, FRAP are not given in line28.
Comment 2: It could be mentioned that the city Pachuca Hidalgo is in Mexico in line98.
Comment 3: Botanical names of Lettuce and Radish should be in Italics in line 103
Comment 4: The botanical name of tomato has changed to Solanum and species is also not mentioned in line 104.
Comment 5: They vegetables should be these vegetables in line 104
Comment 6: In equation,
L, Δa and Δb are not mentioned in line 123.
Comment 7: Acid oxalic should be oxalic acid in line 157.
Comment 8: The name of chemical is 2,6-dichlorophenolindophenol DCPIP in line 159.
Comment 9: The name of scientist is ‘Fernandez -Lopez’ in line 211.
Comment 10: There should be a full stop mark after samples in line 184.
Comment 11: The word ‘planthe t’ does not make sense in line 315.
Comment 12: The word ‘ethe’ also not make sense in line 369.
Comment 13: The spellings of ‘anthocyanins’ are incorrect in table 5.
Comment 14: There should be a space between ‘methodmakes’ in line 426.
Comment 15:The article could be more representative with graphs etc.
Comment 16: Line no 25-26 check the line for grammatic error
Comment 17: Italicize the scientific name
Comment 18: Mention the climatic, field condition, soil condition
Comment 19: Were the crop differentiated on the basis of their growth
Season.
Author Response
Response to Reviewer 2 Comments
We would like to thank reviewers for the comments made to our manuscript (ID: sustainability-2235770) entitled “Comparison of vegetables of ecological and commercial production: physicochemical and antioxidant properties”. Your contributions and suggestions have been considered and we include our responses point-by-point to the comments raised by the reviewers to improve the quality of the manuscript.
Comment 1: Full forms of ABTS, DPPH, FRAP are not given in line 28.
Answer: The full forms has been added in line 25-27
Comment 2: It could be mentioned that the city Pachuca Hidalgo is in Mexico in line 98.
Answer: The country has been mentioned in line 108
Comment 3: Botanical names of Lettuce and Radish should be in Italics in line 103
Answer: The botanical names has been changed to italics in line 120
Comment 4: The botanical name of tomato has changed to Solanum and species is also not mentioned in line 104.
Answer: The species have been mentioned in line 121
Comment 5: They vegetables should be these vegetables in line 104
Answer: The word has been corrected in line 122
Comment 6: In equation,L, Δa and Δb are not mentioned in line 123.
Answer: The equations have been mentioned in lines 141-146
Comment 7: Acid oxalic should be oxalic acid in line 157.
Answer: The name of reactive has been corrected in line 183
Comment 8: The name of chemical is 2,6-dichlorophenolindophenol DCPIP in line 159.
Answer: The chemical name has been corrected in line 185
Comment 9: The name of scientist is ‘Fernandez -Lopez’ in line 211.
Answer: The scientist's name has been corrected in line 209
Comment 10: There should be a full stop mark after samples in line 184.
Answer: The point has been added
Comment 11: The word ‘planthe t’ does not make sense in line 315.
Answer: The word has been corrected in line 377
Comment 12: The word ‘ethe’ also not make sense in line 369.
Answer: The word has been corrected in line 440
Comment 13: The spellings of ‘anthocyanins’ are incorrect in table 5.
Answer: The word has been changed to language English. Besides your suggestion as well as reviewer 1, table 5 was changed by a figure (Figure 1)
Comment 14: There should be a space between ‘methodmakes’ in line 426.
Answer: The space has been added in line 497
Comment 15:The article could be more representative with graphs etc.
Answer: Table 5 has been changed by Figure 1. However, the other tables were not changed to figure due it would be very saturated with bars. Therefore it would not look aesthetic, as well as legible
Comment 16: Line no 25-26 check the line for grammatic error
Answer: At the suggestion of reviewer 1, the abstract has been restructured, line 25-32.
Comment 17: Italicize the scientific name
Answer: The scientific name has been corrected in line 121
Comment 18: Mention the climatic, field condition, soil condition
Answer: The information has been added in lines 108- 112
Comment 19: Were the crop differentiated on the basis of their growth season
Answer: The methodologies of the ecological process have been included in the materials and methods section. However, the production conditions of the commercial sample were not described, because the vegetables were purchased from a commercial distributor and ready for consumption.
It is essential to mention that the objective of the present experiment was not to compare the production systems but compare the quality of the vegetables at the time of purchase for consumption.

Reviewer 3 Report
1. The statistical analysis of repeated samples lacks analysis of variance, which can not accurately describe whether the results of repeated samples are consistent, affecting the reliability of the data.
2. Sample representativeness: The paper repeatedly emphasizes the impact of sample freshness on test indicators (L21-23, L66-68, L88-90, L311-312, L341-345, L374-376, L388-389), so how about the picking time and storage conditions of commercial production obtained from the local supplier of Pachuca Hidalgo, whether they are consistent with ecological products , and how to ensure sample quality? After all, some indicators such as ascorbic acid and intensity decline exponentially with the standing for time. It is suggested that the follow-up test should be carried out in the same test plot to ensure the quality of the sample and thus improve the reliability of the results.
3. The 44th references are not relevant to this paper.
Author Response
Response to Reviewer 3 Comments
We would like to thank reviewers for the comments made to our manuscript (ID: sustainability-2235770) entitled “Comparison of vegetables of ecological and commercial production: physicochemical and antioxidant properties”. Your contributions and suggestions have been considered and we include our responses point-by-point to the comments raised by the reviewers to improve the quality of the manuscript.
Point 1. The authors considered that it was not essential to place the value of the variance of the repetitions because it is a data used to obtain the standard deviation, whose value is reported in the data description and was considered for its validity.
Answer: To obtain the standard deviation, the SPSS statistical program calculates the variance of the repetitions; however, the data is not reported because it is implicit in the standard deviation
Point 2. Sample representativeness: The paper repeatedly emphasizes the impact of sample freshness on test indicators (L21-23, L66-68, L88-90, L311-312, L341-345, L374-376, L388-389), so how about the picking time and storage conditions of commercial production obtained from the local supplier of Pachuca Hidalgo whether they are consistent with ecological products , and how to ensure sample quality? After all, some indicators such as ascorbic acid and intensity decline exponentially with the standing for time.
Answer: Indeed, you are right, the commercial product at the time of purchase has decreased in intensity and ascorbic acid due to time and postharvest handling until it reaches the point of sale. While the ecological sample is harvested and analyzed at the moment the results reflected that they have better color characteristics and bioactive compounds such as ascorbic acid. Therefore, this is an advantage of the vegetables produced in the organic garden for self-consumption, because they can be kept on the plant for a longer time and harvested when they will be used. While the commercial product has a more extensive distribution chain
Point 2.1 It is suggested that the follow-up test should be carried out in the same test plot to ensure the quality of the sample and thus improve the reliability of the results.
Answer: Your suggestion is very interesting, however, this test was not part of the objective of the study, but it will be considered in future studies.
Point 3. The 44th references are not relevant to this paper.
Answer: The reference has been changed

Round 2
Reviewer 1 Report
Dear autors
The authors have made an effort to improve the article but they still need to improve the quality of Figure 1, which does not follow the journal's validity standards. The last paragraph of the conclusions is not very successful. It should be replaced by: However, it is essential to carry out more studies to clarify whether the differences found in favor of organic products are reflected in the health of the people who consume them and thus promote the establishment of urban and peri-urban organic gardens as a strategy of sustainability for healthy eating.
Best regardas
Author Response
Response to Reviewer 1 Comments
We would like to thank reviewers for the comments made to our manuscript (ID: sustainability-2235770) entitled “Comparison of vegetables of ecological and commercial production: physicochemical and antioxidant properties”. Your contributions and suggestions have been considered and we include our responses point-by-point to the comments raised by the reviewers to improve the quality of the manuscript.
Point 1 The authors have made an effort to improve the article but they still need to improve the quality of Figure 1, which does not follow the journal's validity standards.
Answer: The format of Figure 1 has been modified and updated in the document
Point 1.1 The last paragraph of the conclusions is not very successful. It should be replaced by: However, it is essential to carry out more studies to clarify whether the differences found in favor of organic products are reflected in the health of the people who consume them and thus promote the establishment of urban and peri-urban organic gardens as a strategy of sustainability for healthy eating.
Answer: The wording of the last paragraph of the conclusion has been modified according to the suggestion made by the reviewer.